# Relationship between Decreased Mineral Intake Due to Oral Frailty and Bone Mineral Density: Findings from Shika Study

**DOI:** 10.3390/nu13041193

**Published:** 2021-04-05

**Authors:** Fumihiko Suzuki, Shigefumi Okamoto, Sakae Miyagi, Hiromasa Tsujiguchi, Akinori Hara, Thao Thi Thu Nguyen, Yukari Shimizu, Koichiro Hayashi, Keita Suzuki, Shingo Nakai, Masateru Miyagi, Takayuki Kannon, Atsushi Tajima, Hirohito Tsuboi, Tadashi Konoshita, Hiroyuki Nakamura

**Affiliations:** 1Department of Environmental and Preventive Medicine, Graduate School of Medical Science, Kanazawa University, Kanazawa, Ishikawa 920-8640, Japan; f-suzuki@stu.kanazawa-u.ac.jp (F.S.); t-hiromasa@med.kanazawa-u.ac.jp (H.T.); ahara@m-kanazawa.jp (A.H.); toi_fs@yahoo.com (T.T.T.N.); yukari.shimizu@komatsu-u.ac.jp (Y.S.); orihciok1003@gmail.com (K.H.); keitasuzuk@stu.kanazawa-u.ac.jp (K.S.); tenhouwot@gmail.com (S.N.); okitanpopo@yahoo.co.jp (M.M.); 2Community Medicine Support Dentistry, Ohu University Hospital, Koriyama, Fukushima 963-8611, Japan; 3Institute of Medical, Pharmaceutical and Health Sciences, Kanazawa University, Kanazawa, Ishikawa 920-1192, Japan; sokamoto@mhs.mp.kanazawa-u.ac.jp (S.O.); tsuboih@p.kanazawa-u.ac.jp (H.T.); 4Department of Clinical Laboratory Science, Faculty of Health Sciences, Institute of Medical, Pharmaceutical and Health Sciences, Kanazawa University, Kanazawa, Ishikawa 920-0942, Japan; 5Innovative Clinical Research Center, Kanazawa University, Kanazawa, Ishikawa 920-8641, Japan; smiyagi@staff.kanazawa-u.ac.jp; 6Advanced Preventive Medical Sciences Research Center, Kanazawa University, Kanazawa, Ishikawa 920-8640, Japan; kannon@med.kanazawa-u.ac.jp (T.K.); atajima@med.kanazawa-u.ac.jp (A.T.); 7Faculty of Public Health, Haiphong University of Medicine and Pharmacy, Ngo Quyen, Hai Phong 180000, Vietnam; 8Department of Nursing, Faculty of Health Sciences, Komatsu University, Komatsu, Ishikawa 923-0961, Japan; 9Department of Public Health, Graduate School of Advanced Preventive Medical Sciences, Kanazawa University, Kanazawa, Ishikawa 920-8640, Japan; 10Department of Bioinformatics and Genomics, Graduate School of Advanced Preventive Medical Sciences, Kanazawa University, Kanazawa, Ishikawa 920-8640, Japan; 11Third Department of Internal Medicine, Fukui University School of Medicine, Yoshida-gun, Fukui 914-0055, Japan; konosita@u-fukui.ac.jp

**Keywords:** oral frailty, bone mineral density, mineral intake, osteo-sono assessment index

## Abstract

The relationship between oral frailty (OF) and bone mineral density is unclear. This cross-sectional study analyzed the relationship between mineral intake and bone mineral density in middle-aged and older people with pre-oral and OF. The participants, which included 240 people aged 40 years and older, completed the three oral questions on the Kihon Checklist (KCL), which is a self-reported comprehensive health checklist, the brief-type self-administered diet history questionnaire (BDHQ), and the osteo-sono assessment index (OSI). A two-way analysis of covariance on oral function and OSI indicated that the intake of potassium, magnesium, phosphorus, squid/octopus/shrimp/shellfish, carrots/pumpkins, and mushroom was significantly lower in the OF and low-OSI groups than in the non-OF and high-OSI groups. A multiple logistic regression analysis for OF showed that potassium, magnesium, phosphorous and carrots/pumpkins were significantly associated with OF in the low-OSI group but not in the high-OSI group. These results demonstrated that the decrease in mineral intake due to OF was associated with decreased bone mineral density, suggesting that the maintenance of oral function prevents a decrease in bone mineral density.

## 1. Introduction

Oral frailty (OF) has been defined as the accumulation of a slightly poor status in oral conditions and function that is considered a strong prediction of physical frailty [1]. Tanaka et al. [1] reported that OF was significantly associated with 2.4-, 2.2-, 2.3-, and 2.2-fold increased risk of physical frailty, sarcopenia, disability, and mortality, respectively. A previous study [2] demonstrated that the risk of frailty was associated with lower occlusal force, masseter muscle thickness, and oral diadochokinetic rate. There have been several reports on the association between systemic frailty and mineral intake [3,4,5]. A review by Morante et al. [3] reported that dietary factors associated with frailty were calorie, protein, vitamin D, and calcium intake. Moreover, a cohort study [4] demonstrated that low sodium intake (<2504 mg) was associated with frailty in the elderly. Furthermore, a cross-sectional study among middle- and older-aged groups suggested that dietary magnesium intake was also associated with the risk of frailty. However, few studies have evaluated the relationship between OF and mineral intake. A cross-sectional study that investigated the relationship between oral function and nutrient intake found that there was no relationship between oral health behavior (including twice a day tooth brushing or regular attendance of dental clinic) and macro- or micromineral intake [6]. However, the study did not examine systemic factors such as osteoporosis. Due to the importance of systemic factors in bone mineral density, it is necessary to evaluate its influence in the relationship between OF and mineral intake.

With regard to the relationship between bone mineral density and oral cavity, it has been reported that there is a relationship between decreased bone mineral density and the progression of periodontal disease, especially alveolar bone resorption in postmenopausal women [7,8,9]. Grocholewicza and Bohatyrewicz [7] reported a negative correlation between the lumbar bone mineral density and the periodontal disease index and between the radius bone mineral density and the papillary bleeding index. Inagaki et al. [9] also demonstrated that mineral bone density loss of metacarpal was associated with periodontitis and tooth loss in Japanese women. Therefore, osteoporosis has been considered to be a risk factor for periodontal disease. A systematic review by Gerritsen et al. [10] has shown that tooth loss is associated with impairment of oral health-related quality of life. Although it has been reported that systemic frailty is associated with reduced bone mineral density [11], it remains unclear whether OF is directly associated with these reductions.

On the basis of the association between OF, mineral intake, and bone mineral density, it is necessary to examine the interactions between these factors. Therefore, this study aimed to investigate the relationship between mineral intake and bone mineral density in middle-aged and older people with oral dysfunction, including pre-OF.

## 2. Materials and Methods

### 2.1. Study Design and Site

This was a cross-sectional study conducted among residents of Shika Town, Ishikawa Prefecture, Japan, termed the Shika study. As of November 2017, there were 21,007 residents in Shika Town, and 13,713 were older than 40 years [12]. The Shika study epidemiologically investigated the causes of lifestyle-related diseases through interviews, self-administered questionnaires, and comprehensive medical examinations [13,14,15].

### 2.2. Participants

A total of 253 people aged 40 years and older who live in four model districts (Horimatsu, Tsuchida, Higashi Matsudo, and Togi) of Shika Town provided their consent to participate in this study. Of these individuals, 13 were excluded because they did not evaluate osteo-sono assessment index (OSI) or did not have energy records within 600–4000 Kcal on the brief-type self-administered diet history questionnaire (BDHQ)). Figure 1 shows the inclusion criteria. A total of 240 participants who answered all relevant questions in the questionnaires and did not withdraw their consent were included in the analysis.

### 2.3. Questionnaire and Measurements

Comprehensive health survey data were collected from the residents of Shika Town, Ishikawa Prefecture, Japan, between November 2017 and February 2018. The participants completed a self-administered questionnaire on lifestyle and underlying diseases. Lifestyle items included current smokers (1. no, 2. yes) and/or current drinkers (1. no, 2. yes) and education (1. junior high school, 2. high school 3. junior college, 4. university or higher). Underlying disease items included hypertension (1. no, 2. yes), diabetes (1. no, 2. yes), and osteoporosis (1. no, 2. yes). Body mass index (BMI) was measured using health survey data from the Shika study.

The Kihon Checklist (KCL) was used to evaluate OF. KCL is a self-reported comprehensive health checklist designed by a study group from the Ministry of Health, Labour, and Welfare as a screening tool to identify community-dwelling older adults who are vulnerable to frailty and have a higher risk of becoming dependent [16,17]. The validity of the KCL has been demonstrated in previous studies [18,19]. The three oral questions in the KCL were used as components of OF in this study. Specifically, difficulties eating tough foods from half a year ago, difficulties in swallowing tea or soup, and experience of having a dry mouth were defined as the eating, the swallowing, and the oral dryness domain, respectively. For each question, “yes” was converted to 1 point, and the total point was defined as the OF score. Furthermore, as an evaluation of oral function, we asked whether they brushed their teeth at least twice a day and recorded the current number of teeth.

Nutrient intake was assessed using the BDHQ [20,21]. The BDHQ is a four-page structured questionnaire that assesses the consumption frequency of 58 foods and beverages commonly consumed by the general Japanese population. The BDHQ estimates dietary intake in the last month using an ad hoc computer algorithm. The validity of the BDHQ has been demonstrated in previous studies [20,21]. To analyze nutrient data, the density method was used to estimate intake per 1000 Kcal.

Using a quantitative ultrasonic device (AOS-100NW-B, Hitachi Aloka Medical, Tokyo, Japan), the OSI of the right calcaneus was measured as an indicator of bone strength. OSI correlates closely with bone mineral density measured by dual-energy X-ray absorptiometry [22]. OSI was calculated using the following formula:OSI  =  transmission index × speed of sound 2.

### 2.4. Statistical Analysis

The participants were classified into two OF groups: the non-OF group, which had a score of 0, and the OF group, which had a score of 1 or higher. The two OSI groups were classified into the low-OSI and high-OSI group based on the median of the participants in this study. IBM SPSS Statistics version 25 for Windows (IBM, Armonk, NY, USA) was used for the statistical analysis. Student’s t-test was used to determine the association between continuous variables, while the Chi-square test was performed to investigate the association between categorical variables. A two-way analysis of covariance (ANCOVA) was used to examine the effects of the interaction between OF and OSI on mineral and food intake. The following confounding factors were adjusted for age, sex, BMI, current smoker, and current drinker. A multiple logistic regression analysis was conducted to examine the effects of OF and mineral and food intake on bone density, using the OSI as the dependent variable. In addition, the analyses were stratified by OF. Pearson’s correlation coefficient was used to confirm multicollinearity. Specifically, there was no value of |r| > 0.9 in the correlation matrix table between independent variables. The forced input method was used for variable selection. The significance level was set at 5%.

### 2.5. Ethics Statement

This study was conducted with the approval of the Ethics Committee of Kanazawa University (No. 1491). Written informed consent was obtained from all participants prior to participation.

## 3. Results

### 3.1. Participant Characteristics

The participants’ characteristics, OF, OSI, and mineral intakes are shown in Table 1. Of the 240 participants, 125 were males and 115 were females. Participant age ranged from 41 to 86 years, with a mean age of 60.2 ± 10.0 years. There was no significant difference between genders. BMI (*p* < 0.001) and OSI (*p* < 0.001) were significantly higher in males than in females. Significantly more males were current smokers, current drinkers, and better educated and had diabetes than females. On the other hand, osteoporosis was significantly higher among the females. The OF total scores were 0.7 (*SD* = 0.8) in both the male and female groups, with no significant difference between genders. A significantly higher number of females brushed more than twice a day than males. When comparing minerals and foods, the total energy and salt intake were significantly higher in males. Conversely, the intake of minerals, sodium, potassium, calcium, magnesium, phosphorus, green leafy vegetables, carrots/pumpkins, mushroom, and citrus was significantly higher in females.

### 3.2. Comparison of OF

The mean age of the 115 participants in the non-OF group was 57.6 years, which was significantly lower than that of the 125 participants in the OF group (62.5 years) (Table 2). The proportion of participants with eating, swallowing, and oral dryness domain, as well as OF total score and number of teeth, were significantly higher in the OF group. When comparing each nutrient, there was no significant difference between the two groups.

### 3.3. Comparisons with OSI

The mean age of the 106 participants in the low-OSI group (63.0 years) was significantly higher than that of the 134 participants in the high-OSI group (58.0 years, *p* < 0.001) (Table 3). The mean OSI was significantly higher in the high-OSI group. BMI was significantly higher in the high-OSI group than in the low-OSI group. The high-OSI group significantly had a higher number of males, current smokers, current drinkers, and diabetics than the low-OSI group, whereas the low-OSI group had a significantly higher number of people with osteoporosis. When comparing each nutrient, the intake of minerals, sodium, potassium, calcium, magnesium, phosphorus, and citrus was significantly higher in the low-OSI group than in the high-OSI group, whereas that of total energy was significantly higher in the high-OSI group.

### 3.4. Main Effects and Interaction between OF and the OSI Groups

When the non-OF group was subdivided into two groups based on OSI, there were 50 participants in the low-OSI group and 65 in the high-OSI group. The subdivision of the OF group into two groups based on the OSI similarly resulted into 56 participants in the low-OSI group and 69 in the high-OSI group (Table 4). A two-way ANCOVA adjusting for age, sex, BMI, current smoker, and current drinker was used to examine the main effects and interactions between OF and the OSI on minerals and food intake. The main effect was observed for potassium (*p* = 0.013), magnesium (*p* = 0.044), phosphorus (*p* = 0.047), green leafy vegetables (*p* = 0.007), citrus (*p* = 0.029), and salt (*p* = 0.043) intake between the two OF groups, but not between two OSI groups. Interactions were observed for sex (*p* = 0.037), BMI (*p* = 0.013), number of teeth (*p* = 0.003), potassium (*p* = 0.038), magnesium (*p* = 0.031), phosphorus (*p* = 0.022), squid/octopus/shrimp/selfish (*p* = 0.016), carrots/pumpkins (*p* = 0.044), and mushroom (*p* = 0.045). A post hoc Bonferroni analysis indicated a significantly lower intake of potassium (*p* = 0.002), magnesium (*p* = 0.005), and phosphorus (*p* = 0.004) in the OF group than in the non-OF group with low OSI (Appendix A). Similarly, there was a significantly lower intake of squid/octopus/shrimp/selfish (*p* = 0.029) and carrots/pumpkins (*p* = 0.037) in the low- OSI group than in the high-OSI group with OF (Appendix A). Table 5 shows the correlation between the minerals (potassium, magnesium, and phosphorus) and food (squid/octopus/shrimp/selfish, green leafy vegetables, carrots/pumpkins, mushroom, citrus, and salt), which indicates the main effect or the interaction in Table 4. Significant correlations between all combinations of minerals and foods, except squid/octopus/shrimp/selfish and potassium, were observed. This correlation demonstrated that many of the minerals and foods interacted with OF and OSI, suggesting that the decrease in bone density is related to the decrease in mineral-containing foods that are difficult to chew caused by OF.

### 3.5. Effects of OF and Mineral and Food Intake on OSI

Table 6 shows the results of multiple logistic regression analysis stratified by OSI with OF as the dependent variable and mineral intake as the independent variable. Minerals (OR: 0.767; 95% CI: 0.621, 0.948; *p* = 0.014), potassium (OR: 0.998; 95% CI: 0.997, 0.999; *p* = 0.004), calcium (OR: 0.995; 95% CI: 0.991, 0.999; *p* = 0.028), magnesium (OR: 0.981; 95% CI: 0.968, 0.994; *p* = 0.006), and phosphorus (OR: 0.995; 95% CI: 0.991, 0.998; *p* = 0.004) were significant independent variables in the model with adjustment for age, sex, BMI, current smoker, and current drinker in the low-OSI group but not in the high-OSI group. This result implies that bone density is negatively associated with low mineral intake only in the low-OSI group with OF. Table 7 shows the results of the same analysis and confounding factor adjustment but with food as the independent variable. Carrots/pumpkins (OR: 0.971; 95% CI: 0.944, 1.000; *p* = 0.047), citrus (OR: 0.986; 95% CI: 0.974, 0.999; *p* = 0.038), and salt (OR: 0.601; 95% CI: 0.410, 0.882; *p* = 0.009) were significant independent variables in the model in the low-OSI group but not in the high-OSI group. This result implies that bone density is negatively associated with low carrots/pumpkins, citrus, and salt intake only in the low-OSI group with OF.

## 4. Discussion

Since oral frailty is a relatively new concept, its evaluation method is not well established. In the long-term care insurance system for the elderly in Japan, three domains related to the oral cavity of the KCL are used for the evaluation items of oral dysfunction [18,19]. In addition to these three domains, the current number of teeth [1,23], masticatory ability [1], tooth brushing twice daily [6], regular dental visits [6], denture use [24], occlusal force [2], tongue–lip function [2], and eating alone [25] were used to evaluate OF. Furthermore, since the evaluation method of pre-OF has not been fully discussed, there have been some reports [25,26] that the survey items of OF with few applicable items are regarded as pre-OF. In this study, we decided to use three domains related to the oral cavity in the KCL to evaluate the oral dysfunction from an early stage, including pre-OF, as a population approach.

The main result of this study was that when OSI was stratified into two groups with OF as the dependent variable in multiple logistic regression analysis, minerals, potassium, calcium, magnesium, phosphorus, carrots/pumpkins, citrus, and salt intake were found to be significant independent variables in the low-OSI group.

According to ANCOVA results, the main effects were observed in potassium, magnesium, phosphorus, green leafy vegetables, citrus, and salt intake even after adjusting for confounding factors in the OF group. This contrasted the study by Nomura et al. [6], who investigated the relationship between OF and nutrient intake and found no association between OF and mineral intake. They evaluated factors related to oral health behavior and macronutrients using structural equation modeling. It seems that the reason why our results differed from theirs was that they did not analyze the involvement of chewing and swallowing functions in the assessment of food choice and oral dysfunction. Alternatively, Zhe et al. [23] reported that protein, vitamins, and mineral intake is positively associated with the total number of natural teeth. This study was thought to have elucidated the non-oral frailty and high-OSI group had 20 or more current teeth, while the other groups had less than 20 current teeth, indicating an interaction. In other words, it indicates that there is a relationship between the decline in oral function and food choice.

It has also been noted that not nutrients or foods, but HbA1c, diabetes, and currently drunk showed a main effect on the OSI even after adjusting the covariates. A meta-analysis by Ma et al. [27] and a review by Piepkorn et al. [28] reported that patients with type 2 diabetes have higher bone mineral density than non-diabetic patients. Our results that showed that the proportion of patients with diabetes was higher in the high-OSI group support this finding. A prospective study by Holbrook and Barrett-Connor [29] also reported that drinking increased bone mineral density. Our findings that the proportion of current drinkers was higher in the high-OSI group might also support the report.

Potassium, magnesium, phosphorus, carrots/pumpkins, citrus, and salt intake were observed to have the main effect on the two OF groups or the interaction among the four OF and OSI groups. These minerals and foods were also significant independent variables when stratified by OSI in multiple logistic regression analysis with OF as the dependent variable. In line with our finding, previous studies investigating the relationship between mineral intake and bone mineral density reported that potassium and magnesium intake was associated with increased bone mineral density in the elderly [30]; that dietary protein, phosphorus, and potassium were beneficial to bone mineral density in adult men consuming adequate dietary calcium [31]; and that magnesium intake was associated with bone mineral density [5,32]. With regard to the relationship between food intake and bone mineral density, Kim et al. [33] reported that intake of anchovies, radishes, carrots, zucchinis, and tomatoes were significant important factors in minimizing bone density risk. Our results support the above findings because we found a similar relationship between the minerals or mineral-containing food intake and OSI. Our study showed that the mean current number of teeth was 20 or more only in the non-OF and high-OSI group. Poor chewing and swallowing function due to OF and having less than 20 current teeth are presumed to reduce the variety of chewable foods. It suggests that the decrease in mineral intake due to the change in the food choice by OF is associated with the decrease in bone mineral density.

Since this study was a cross-sectional study, the causal relationship between OF, mineral intake, and bone mineral density could not be ascertained. Moreover, OSI does not directly evaluate bone mineral density. Finally, self-administration of BDHQ may lack objectivity.

## 5. Conclusions

Our study demonstrated a decrease in mineral intake due to OF, which was associated with decreased bone mineral density. These results suggest that the maintenance of oral function prevents decrease in bone mineral density.

## Figures and Tables

**Figure 1 nutrients-13-01193-f001:**
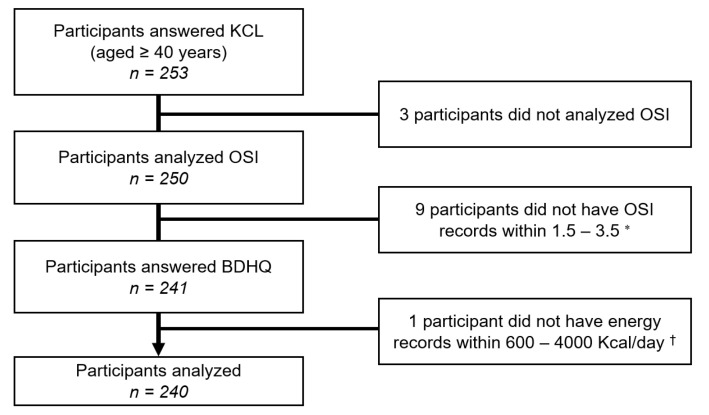
Participant recruitment chart. * This value was set to the OSI mean ± 2SD. † This reference value was chosen for the following reasons: less than 600 kcal/day is equivalent to half the energy intake required for the lowest physical activity category, and more than 4000 kcal/day is equivalent to 1.5 times the energy intake required for the medium physical activity category. Abbreviations: KCL, the Kihon checklist, OSI, the osteo-sono assessment index, BDHQ, the brief-type self-administered diet history questionnaire.

**Table 1 nutrients-13-01193-t001:** Participant characteristics.

	Total (*N* = 240)		Male (*n* = 125)		Female (*n* = 115)		*p*-Value *
	Mean (*n*)	*SD* (%)	Mean (*n*)	*SD* (%)	Mean (*n*)	*SD* (%)	
Age, years	60.2	10.0	59.6	9.2	60.8	10.8	0.677
BMI, kg/m^2^	23.2	3.1	23.9	3.2	22.5	2.9	**<0.001**
Current smoker, *n* (%)	124	51.7	93	74.4	31	27.0	**<0.001**
Current drinker, *n* (%)	48	20.0	41	32.8	7	6.1	**<0.001**
Education							**0.020**
Junior high school, *n* (%)	34	14.2	15	12.0	19	16.5	
High school, *n* (%)	67	27.9	36	28.8	31	27.0	
Junior college, *n* (%)	37	15.4	14	11.2	23	20.0	
University or higher, *n* (%)	28	11.7	21	16.8	7	6.1	
Hypertension, *n* (%)	78	32.5	46	36.8	32	27.8	0.162
Diabetes, *n* (%)	17	7.1	13	10.4	4	3.5	**0.045**
Osteoporosis, *n* (%)	9	3.8	1	0.8	8	7.0	**0.015**
Eating domain, *n* (%)	68	28.3	41	32.8	27	23.5	0.117
Swallowing domain, *n* (%)	53	22.1	25	20.0	28	24.3	0.440
Oral dryness domain, *n* (%)	44	18.3	22	17.6	22	19.1	0.868
Oral frailty (OF), total score	0.7	0.8	0.7	0.8	0.7	0.8	0.604
Number of teeth, *n* (%)	18.9	9.4	18.8	9.0	19.0	9.8	0.886
More than twice a day brushing, *n* (%)	50	20.8	15	12.0	35	30.4	**0.001**
OSI	2.7	0.3	2.8	0.3	2.5	0.3	**<0.001**
Total energy, Kcal	1927.53	606.45	2128.63	627.81	1708.95	499.80	**<0.001**
Minerals, % energy	10.21	2.00	9.57	1.84	10.90	1.95	**<0.001**
Sodium, mg/1000 Kcal	2386.24	501.75	2336.33	500.57	2440.49	499.55	**<0.001**
Potassium, mg/1000 Kcal	1383.74	425.68	1195.40	312.30	1588.46	438.90	**<0.001**
Calcium, mg/1000 Kcal	287.38	106.20	251.89	95.40	325.96	104.21	**<0.001**
Magnesium, mg/1000 Kcal	138.82	31.52	126.80	25.62	151.87	32.22	**<0.001**
Phosphorus, mg/1000 Kcal	566.36	120.94	523.63	109.99	612.80	115.46	**<0.001**
Squid/octopus/shrimp/shellfish, g/1000 Kcal	16.56	15.73	17.85	17.02	15.16	14.13	0.186
Green leafy vegetables, g/1000 Kcal	40.08	40.00	33.26	35.48	47.49	43.34	**0.006**
Carrots/pumpkins, g/1000 Kcal	22.06	16.23	19.17	14.94	25.19	17.05	**0.004**
Mushroom, g/1000 Kcal	11.75	9.50	10.01	8.61	13.65	10.08	**0.003**
Citrus, g/1000 Kcal	21.49	28.45	16.52	21.85	26.89	33.49	**0.005**
Salt, g/1000 Kcal	3.41	1.19	3.58	1.25	3.22	1.10	**0.020**

* *p*-values were calculated using Student’s *t*-tests and Chi-square test for continuous and categorical variables, respectively (*p*-values less than 0.05 are highlighted in bold). Abbreviations: SD, standard deviation; BMI, body mass index; OSI, osteo-sono assessment index.

**Table 2 nutrients-13-01193-t002:** Differences in characteristics and daily nutrient intake between the OF groups.

	Total (*N* = 240)				
	Non-OF (*n* = 115)		OF (*n* = 125)		*p*-Value *
	Mean (*n*)	*SD* (%)	Mean (*n*)	*SD* (%)	
Age, years	57.6	9.2	62.5	10.2	**<0.001**
Sex (male), *n* (%)	59	51.3	66	52.8	0.897
BMI, kg/m^2^	23.4	3.2	23.1	3.1	0.354
Current smoker, *n* (%)	58	50.4	66	52.8	0.796
Current drinker, *n* (%)	25	21.7	23	18.4	0.524
Hypertension, *n* (%)	32	27.8	46	36.8	0.159
Diabetes, *n* (%)	11	9.6	6	4.8	0.207
Osteoporosis, *n* (%)	2	1.7	7	5.6	0.174
Eating domain, *n* (%)	0	0.0	68	54.4	**<0.001**
Swallowing domain, *n* (%)	0	0.0	53	42.4	**<0.001**
Oral dryness domain, *n* (%)	0	0.0	44	35.2	**<0.001**
OF, total score	0.0	0.0	1.3	0.5	**<0.001**
Number of teeth, *n* (%)	21.1	8.7	16.8	9.6	**<0.001**
More than twice a day brushing, *n* (%)	24.0	20.9	26.0	20.8	1.000
OSI	2.7	0.3	2.7	0.3	0.566
Total energy, Kcal	1945.59	628.51	1910.92	587.47	0.659
Minerals, % energy	10.29	2.17	10.13	1.84	0.531
Sodium, mg/1000 Kcal	2394.40	542.01	2378.73	463.73	0.810
Potassium, mg/1000 Kcal	1412.76	436.21	1357.04	415.71	0.312
Calcium, mg/1000 Kcal	287.49	103.61	287.28	108.94	0.988
Magnesium, mg/1000 Kcal	140.10	32.61	137.63	30.56	0.545
Phosphorus, mg/1000 Kcal	571.74	122.41	561.41	119.86	0.510
Squid/octopus/shrimp/shellfish, g/1000 Kcal	16.74	15.98	16.39	15.55	0.864
Green leafy vegetables, g/1000 Kcal	42.98	44.00	37.41	35.90	0.282
Carrots/pumpkins, g/1000 Kcal	21.68	14.48	22.41	17.74	0.728
Mushroom, g/1000 Kcal	11.75	9.76	11.75	9.29	1.000
Citrus, g/1000 Kcal	24.23	33.21	18.96	23.09	0.158
Salt, g/1000 Kcal	3.49	1.23	3.33	1.16	0.301

* *p*-values were calculated using Student’s *t*-tests and Chi-square test for continuous and categorical variables, respectively (*p*-values less than 0.05 are highlighted in bold). Abbreviation: OF, oral frailty.

**Table 3 nutrients-13-01193-t003:** Differences in characteristics and daily nutrient intake between the OSI groups.

	Total (*N* = 240)				
	Low OSI (*n* = 106)		High OSI (*n* = 134)		*p*-Value *
	Mean (*n*)	*SD* (%)	Mean (*n*)	*SD* (%)	
Age, years	63.0	10.5	58.0	9.1	**<0.001**
Sex (male), *n* (%)	29	27.4	96	71.6	**<0.001**
BMI, kg/m^2^	22.4	2.7	23.9	3.3	**<0.001**
Current smoker, *n* (%)	35	33.0	89	66.4	**<0.001**
Current drinker, *n* (%)	9	8.5	39	29.1	**<0.001**
Hypertension, *n* (%)	32	30.2	46	34.3	0.671
Diabetes, *n* (%)	2	1.9	15	11.2	**0.009**
Osteoporosis, *n* (%)	8	7.5	1	0.7	**0.011**
Eating domain, *n* (%)	27	25.5	41	30.6	0.392
Swallowing domain, *n* (%)	29	27.4	24	17.9	0.087
Oral dryness domain, *n* (%)	25	23.6	19	14.2	0.067
OF, total score	0.8	0.9	0.6	0.7	0.183
Number of teeth, *n* (%)	18.0	9.3	19.6	9.4	0.180
More than twice a day brushing, *n* (%)	23.0	21.7	27.0	20.1	0.873
OSI	2.4	0.2	2.9	0.2	**<0.001**
Total energy, Kcal	1835.21	583.73	2000.56	616.23	0.036
Minerals, % energy	10.80	2.07	9.73	1.82	**<0.001**
Sodium, mg/1000 Kcal	2467.51	526.16	2321.95	473.73	**0.025**
Potassium, mg/1000 Kcal	1521.23	445.97	1274.98	376.32	**<0.001**
Calcium, mg/1000 Kcal	319.98	115.82	261.60	90.33	**<0.001**
Magnesium, mg/1000 Kcal	147.99	33.45	131.56	27.95	**<0.001**
Phosphorus, mg/1000 Kcal	604.68	125.66	536.05	108.27	**<0.001**
Squid/octopus/shrimp/shellfish, g/1000 Kcal	16.92	17.01	16.27	14.70	0.750
Green leafy vegetables, g/1000 Kcal	43.95	41.26	37.02	38.86	0.183
Carrots/pumpkins, g/1000 Kcal	23.27	15.81	21.10	16.56	0.305
Mushroom, g/1000 Kcal	12.60	10.32	11.08	8.77	0.221
Citrus, g/1000 Kcal	27.65	35.81	16.62	19.71	**0.005**
Salt, g/1000 Kcal	3.37	1.17	3.44	1.21	0.639

* *p*-values were calculated using Student’s *t*-tests and Chi-square test for continuous and categorical variables, respectively (*p*-values less than 0.05 are highlighted in bold). Abbreviation: OSI, the osteo-sono assessment index.

**Table 4 nutrients-13-01193-t004:** Two-way ANCOVA on OF and OSI groups.

	Total (*N* = 240)				*p*-Value *		
	Non-OF (*n* = 115)		OF (*n* = 125)				
	Low-OSI (*n* = 50)	High-OSI (*n* = 65)	Low-OSI (*n* = 56)	High-OSI (*n* = 69)	OF	OSI	OF × OSI
	EMM (95% CI)	EMM (95% CI)	EMM (95% CI)	EMM (95% CI)			
Age, years	59.7 (57.0, 62.4)	56.1 (53.7, 58.5)	65.8 (63.1, 68.4)	59.8 (57.4, 62.1)	**<0.001**	**0.001**	0.348
Sex ^†^	1.4 (1.3, 1.6)	1.6 (1.5, 1.7)	1.3 (1.2, 1.4)	1.7 (1.6, 1.8)	0.858	**<0.001**	**0.037**
BMI, kg/m^2^	22.2 (21.4, 23.1)	24.4 (23.6, 25.2)	22.9 (22.0, 23.8)	23.1 (22.4, 23.9)	0.475	**0.008**	**0.013**
Current smoker ^‡^	1.1 (1.0, 1.2)	1.3 (1.2, 1.4)	1.1 (1.0, 1.2)	1.2 (1.1, 1.3)	0.722	0.062	0.751
Current drinker ^‡^	1.4 (1.3, 1.6)	1.5 (1.4, 1.6)	1.4 (1.3, 1.6)	1.6 (1.5, 1.7)	0.525	**0.047**	0.412
Hypertension ^‡^	1.3 (1.2, 1.4)	1.3 (1.2, 1.5)	1.4 (1.3, 1.6)	1.4 (1.2, 1.5)	0.283	0.896	0.461
Diabetes ^‡^	1.0 (0.9, 1.1)	1.2 (1.1, 1.2)	1.0 (0.9, 1.1)	1.1 (1.0, 1.1)	0.173	**0.013**	0.339
Osteoporosis ^‡^	1.0 (1.0, 1.1)	1.0 (1.0, 1.1)	1.1 (1.0, 1.1)	1.0 (1.0, 1.1)	0.476	0.360	0.294
Number of teeth, *n*	18.4 (16.0, 20.8)	21.6 (19.4, 23.8)	19.7 (17.2, 22.2)	16.1 (14.0, 18.2)	0.066	0.885	**0.003**
More than twice a day brushing, *n*	1.2 (1.0, 1.3)	1.2 (1.1, 1.3)	1.1 (1.0, 1.3)	1.3 (1.2, 1.4)	0.785	0.066	0.691
Total energy, Kcal	1997.15 (1834.08, 2160.22)	1952.80 (1805.82, 2099.78)	1873.01 (1707.91, 2038.10)	1897.53 (1755.37, 2039.70)	0.241	0.910	0.650
Minerals, % energy	10.81 (10.29, 11.32)	10.12 (9.65, 10.58)	9.90 (9.37, 10.42)	10.11 (9.66, 10.56)	0.060	0.392	0.062
Sodium, mg/1000 Kcal	2492.42 (2349.10, 2635.74)	2349.93 (2220.75, 2479.11)	2372.32 (2227.22, 2517.42)	2354.80 (2229.86, 2479.74)	0.391	0.298	0.349
Potassium, mg/1000 Kcal	1500.25 (1397.99, 1602.50)	1393.30 (1301.13, 1485.47)	1281.69 (1178.16, 1385.22)	1373.14 (1284.00, 1462.29)	**0.013**	0.888	**0.038**
Calcium, mg/1000 Kcal	311.00 (285.15, 336.85)	287.24 (263.94, 310.54)	268.05 (241.88, 294.22)	286.11 (263.57, 308.64)	0.070	0.837	0.083
Magnesium, mg/1000 Kcal	146.78 (138.89, 154.66)	138.71 (131.60, 145.81)	131.36 (123.37, 139.34)	139.20 (132.33, 146.08)	**0.044**	0.979	**0.031**
Phosphorus, mg/1000 Kcal	604.90 (574.40, 635.40)	559.33 (531.83, 586.82)	543.59 (512.70, 574.47)	563.54 (536.95, 590.14)	**0.047**	0.434	**0.022**
Squid/octopus/shrimp/shellfish, g/1000 Kcal	21.14 (16.70, 25.57)	14.29 (10.29, 18.29)	14.18 (9.68, 18.67)	17.32 (13.45, 21.19)	0.345	0.437	**0.016**
Green leafy vegetables, g/1000 Kcal	42.22 (31.62, 52.82)	50.90 (41.35, 60.45)	28.40 (17.67, 39.13)	37.81 (28.57, 47.05)	**0.007**	0.112	0.941
Carrots/pumpkins, g/1000 Kcal	23.49 (18.97, 28.01)	21.80 (17.72, 25.87)	17.86 (13.29, 22.44)	24.67 (20.73, 28.61)	0.515	0.292	**0.044**
Mushroom, g/1000 Kcal	12.79 (10.14, 15.44)	11.80 (9.41, 14.19)	9.11 (6.43, 11.80)	13.10 (10.78, 15.41)	0.339	0.293	**0.045**
Citrus, g/1000 Kcal	32.57 (24.70, 40.44)	19.64 (12.54, 26.73)	17.90 (9.93, 25.87)	18.12 (11.26, 24.98)	**0.029**	0.133	0.073
Salt, g/1000 Kcal	3.74 (3.40, 4.07)	3.42 (3.13, 3.72)	3.15 (2.82, 3.49)	3.38 (3.09, 3.67)	**0.043**	0.806	0.083

* Two-way ANCOVA (*p*-values less than 0.05 are highlighted in bold). Adjusted for age, sex, BMI, current smoker, and current drinker. ^†^ (1. female, 2. male), ^‡^ (1. no, 2. yes). Abbreviations: Two-way ANCOVA, two-way analysis of covariance, EMM, estimated marginal means, CI, confidence interval.

**Table 5 nutrients-13-01193-t005:** Correlation between minerals and foods.

	Squid/Octopus/Shrimp/Shellfish, g/1000 Kcal	Green Leafy Vegetables, g/1000 Kcal	Carrots/Pumpkins, g/1000 Kcal	Mushroom, g/1000 Kcal	Citrus, g/1000 Kcal	Salt, g/1000 Kcal
Minerals, % energy	0.226 **(< 0.001)** *	0.415 **(<0.001)**	0.265 **(<0.001)**	0.325 **(<0.001)**	0.194 **(0.003)**	0.239 **(<0.001)**
Sodium, mg/1000 Kcal	0.226 **(<0.001)**	0.105 (0.105)	-0.112 (0.085)	0.070 (0.279)	0.041 (0.527)	**0.192 (0.003)**
Potassium, mg/1000 Kcal	0.123 (0.057)	0.618 **(<0.001)**	0.604 **(<0.001)**	0.478 **(<0.001)**	0.305 **(<0.001)**	0.184 **(0.004)**
Calcium, mg/1000 Kcal	0.125 (0.053)	0.486 **(<0.001)**	0.400 **(<0.001)**	0.375 **(<0.001)**	0.225 **(<0.001)**	0.177 **(0.006)**
Magnesium, mg/1000 Kcal	0.226 **(<0.001)**	0.590 **(<0.001)**	0.521 **(<0.001)**	0.472 **(<0.001)**	0.277 **(<0.001)**	0.232 **(<0.001)**
Phosphorus, mg/1000 Kcal	0.289 **(<0.001)**	0.434 **(<0.001)**	0.416 **(<0.001)**	0.437 **(<0.001)**	0.235 **(<0.001)**	0.384 **(<0.001)**

* Pearson’s correlation coefficient (*p*-value). *p*-values less than 0.05 are highlighted in bold.

**Table 6 nutrients-13-01193-t006:** Relationship between mineral intake and OF stratified by OSI.

		β	*p*-Value	OR	95 % CI	
					Lower	Upper
Low-OSI group	Minerals	−0.265	**0.014**	0.767	0.621	0.948
	Sodium	−0.001	0.193	0.999	0.999	1.000
	Potassium	−0.002	**0.004**	0.998	0.997	0.999
	Calcium	−0.005	**0.028**	0.995	0.991	0.999
	Magnesium	−0.019	**0.006**	0.981	0.968	0.994
	Phosphorus	−0.005	**0.004**	0.995	0.991	0.998
High-OSI group	Minerals	−0.014	0.890	0.986	0.812	1.199
	Sodium	0.000	0.909	1.000	0.999	1.001
	Potassium	0.000	0.747	1.000	0.999	1.001
	Calcium	0.000	0.990	1.000	0.996	1.004
	Magnesium	0.000	0.991	1.000	0.987	1.013
	Phosphorus	0.000	0.960	1.000	0.997	1.003

Significant estimates are in bold. Adjusted for age, sex, body mass index (BMI), current smoker, and current drinker. Abbreviations: OR, odds ratio.

**Table 7 nutrients-13-01193-t007:** Relationship between food intake and OF stratified by OSI.

		β	*p*-Value	OR	95 % CI	
					Lower	Upper
Low-OSI group	Squid/octopus/shrimp/shellfish	−0.027	0.059	0.973	0.946	1.001
	Green leafy vegetables	−0.010	0.078	0.990	0.979	1.001
	Carrots/pumpkins	−0.029	**0.047**	0.971	0.944	1.000
	Mushroom	−0.042	0.052	0.959	0.919	1.000
	Citrus	−0.014	**0.038**	0.986	0.974	0.999
	Salt	−0.509	**0.009**	0.601	0.410	0.882
High-OSI group	Squid/octopus/shrimp/shellfish	0.017	0.189	1.018	0.991	1.045
	Green leafy vegetables	−0.009	0.087	0.991	0.981	1.001
	Carrots/pumpkins	0.014	0.232	1.015	0.991	1.039
	Mushroom	0.015	0.496	1.015	0.972	1.059
	Citrus	−0.005	0.616	0.995	0.976	1.015
	Salt	−0.045	0.784	0.956	0.696	1.315

Significant estimates are in bold. Adjusted for age, sex, BMI, current smoker, and current drinker.

## Data Availability

Available on request and by approval (Kanazawa University Ethics Committee. Person in charge: Yuko Katsuragi email: pub-jim2@staff.kanazawa-u.ac.jp.

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
