# Peer review of "Relationship between Decreased Mineral Intake Due to Oral Frailty and Bone Mineral Density: Findings from Shika Study"

_nutrients, 2021, doi:10.3390/nu13041193_

Round 1

Reviewer 1 Report

INTRODUCTION

The introduction provides a good, generalized background of the topic that quickly gives the reader an appreciation of the wide range of applications and an understanding of the research question.

I think the motivations for this study is more than clear.

Objectives

The objective is clearly defined.

METHODS

The experimental approach is appropriate for the aim of the study.

This section is well described and allows to replicate the study.

RESULTS

The results paragraph include more relevant and extended data.

All of the tables include specific, well-developed statistic.

DISCUSSION

There are some sentences that might be too long and this could make understanding difficult.

The conclusions are according to the discussion, well established and it has coherence with the initial aim.

LITERATURE CITED

The literature cited is relevant to the study.

Reviewer 2 Report

I reviewed the article entitled "Relationship between decreased mineral intake due to oral frailty and bone mineral density: Findings from Shika study" by Fumihiko Suzuki et al. This is a cross-sectional study investigating the correlation of oral frailty with mineral intake and bone mineral density. They have reported that oral frailty is associated with decrease in mineral intake and bone mineral density.

The manuscript is well written but needs edition.

Comments:

  1. Table 2, shows that there is no difference between oral frailty (OF) group and non-oral frailty (non-OF) group in terms of OSI, energy intake, mineral intake (calcium and magnesium). In addition, higher mineral intake is helpful for bone health but low OSI group had higher mineral intake. However, the author concluded that decrease in bone density is related to less mineral intake and OF. How this happens when there is no difference in mineral intake between OF and non-OF groups.
  2. Smoking and drinking alcohol have detrimental effects on bone but table 3, shows the high OSI group has higher number of smoker and drinker, which technically changes the outcomes.
  3. Table 4 needs edition. They can separate the data in different tables to be able to assess the results better or highlight the most important results in the table legend. The current table is difficult to follow and P-value of each comparison should be close to compared items.
  4. They can draw a diagram to visualize the results better
  5. If decrease in mineral intake due to OF is associated with decreased bone mineral density, then participants with OF should have lower bone density and mineral intake.

Bone structure and geometry in young men: the influence of smoking, alcohol intake and physical activity. Kyriacos I Eleftheriou, Jaikirty S Rawal, Lawrence E James, John R Payne, Mike Loosemore, Dudley J Pennell, Michael World, Fotios Drenos, Fares S Haddad, Steve E Humphries, Julie Sanders, Hugh E Montgomery. Bone. 2013 Jan;52(1):17-26.

Reviewer 3 Report

This manuscript analyzed the relationship between mineral intake and bone mineral density in middle-aged and older people with pre-oral and oral frailty (OF) based on the assessment of the Kihon Checklist, BDHQ, and the osteo-sono assessment index (OSI). The authors' analysis demonstrated that the decrease in mineral intake due to OF was associated with decreased bone mineral density, suggesting that the maintenance of oral function is one of the important things to prevent the decrease in bone mineral density.
Comments:
1. Four groups in this manuscript to analyze, the OF-Low OSI, the OF-high OSI, Non-OF-Low OSI, Non-OF-high OSI. In my opinion, the authors did not clearly clarify the relationship between the various groups. 
2. Are the results in this manuscript just indicated that “bone density is negatively associated with low mineral intake only in the low-OSI group with OF”?  What is the association between the other three groups (the high-OSI with OF, low-OSI, or High-OSI with non-OF groups)? 
3. In Table 4, are the numbers shown that the mineral intake in OF Low OSI group is higher than the OF high OSI group? for examples, the potassium (1480.15 in OF-low OSI, 1257.13 in OF-high OSI), magnesium (144.98 in OF-low OSI, 131.67 in OF-high OSI), Phosphorus (591.6 in OF-low OSI, 536.91 in OF-high OSI), green leafy vegetables (42.66 in OF-low OSI, 33.15 in OF-high OSI), only Squid/.. is low in  OF-low OSI (14.21) than in OF-high OSI (18.17)?

4. In table 4, Minerals, % energy line, need a comma between 9.34, and 10.12
5. In table 5, need a space in magnesium, phosphorus line, and Salt column.

Suggestion:
Please clarify the relationship of bone mineral density between OF with Non-OF, and the relationship of mineral intake between OF-low OSI and OF-high OSI.
